# Redox Regulation of LAT Enhances T Cell-Mediated Inflammation

**DOI:** 10.3390/antiox13040499

**Published:** 2024-04-22

**Authors:** Jaime James, Ana Coelho, Gonzalo Fernandez Lahore, Clara M. Hernandez, Florian Forster, Bernard Malissen, Rikard Holmdahl

**Affiliations:** 1Medical Inflammation Research, Division of Immunology, Department of Medical Biochemistry and Biophysics, Karolinska Institutet, 17177 Stockholm, Sweden; jja@immudex.com (J.J.); ana.coelho@ki.se (A.C.); gfernandezlahore@bwh.harvard.edu (G.F.L.); clara.marquina1@monash.edu (C.M.H.); f.forster@sciotec.at (F.F.); 2Centre d’Immunophénomique, Aix Marseille Université, INSERM, 13288 Marseille, France; bernardm@ciml.univ-mrs.fr

**Keywords:** NCF1, reactive oxygen species, linker for activation of T cells, T cell receptor signaling, rheumatoid arthritis

## Abstract

The positional cloning of single nucleotide polymorphisms (SNPs) of the neutrophil cytosolic factor 1 (*Ncf1*) gene, advocating that a low oxidative burst drives autoimmune disease, demands an understanding of the underlying molecular causes. A cellular target could be T cells, which have been shown to be regulated by reactive oxygen species (ROS). However, the pathways by which ROS mediate T cell signaling remain unclear. The adaptor molecule linker for activation of T cells (LAT) is essential for coupling T cell receptor-mediated antigen recognition to downstream responses, and it contains several cysteine residues that have previously been suggested to be involved in redox regulation. To address the possibility that ROS regulate T cell-dependent inflammation through LAT, we established a mouse strain with cysteine-to-serine mutations at positions 120 and 172 (LAT^SS^). We found that redox regulation of LAT through C120 and C172 mediate its localization and phosphorylation. LAT^SS^ mice had reduced numbers of double-positive thymocytes and naïve peripheral T cells. Importantly, redox insensitivity of LAT enhanced T cell-dependent autoimmune inflammation in collagen-induced arthritis (CIA), a mouse model of rheumatoid arthritis (RA). This effect was reversed on an NCF1-mutated (NCF1^m1j^), ROS-deficient, background. Overall, our data show that LAT is redox-regulated, acts to repress T cell activation, and is targeted by ROS induced by NCF1 in antigen-presenting cells (APCs).

## 1. Introduction

The perspective on redox regulation of T cell-mediated autoimmune diseases has shifted after the positional cloning of NCF1 SNPs in both experimental animal models and humans [1,2,3,4]. NCF1 (also named p47phox) is a subunit of the NOX2 complex, the key inducible oxidase in cellular membranes [5]. Further genetic studies in mice, rats, and humans have confirmed that NCF1 mutations promote the development of autoimmune arthritis and lupus by lowering the production of ROS [1,2,3,4,6]. The mechanism whereby ROS regulates inflammation is likely to be very complex as a multitude of pathways are redox-regulated. A pathway suggested to regulate autoimmune arthritis is NCF1-induced ROS by antigen-presenting cells, which could regulate the T cell receptor (TCR) signaling in interacting T cells [7,8]. Upon ligation of the TCR/CD3 complex, the Src family kinase LCK phosphorylates immunoreceptor tyrosine-based activation motifs (ITAMs) on the cytoplasmic tail of CD3 thereby creating a docking site for the kinase ZAP70 [9,10]. Activated ZAP70 phosphorylates LAT (ENSMUSG00000030742) [11], which then acts as an adaptor protein to facilitate the assembly of multiprotein complexes termed LAT signalosomes. These are essential for potentiating and diversifying the initial TCR signal, resulting in the activation of multiple signaling pathways involved in the activation, differentiation, and proliferation of T cells [12,13,14]. The importance of LAT for TCR signaling is demonstrated in LAT-knockout mice, which show a block in T cell development at the double-negative stage and have no mature αβ T cells in the periphery [14,15]. LAT expression, function, and localization have previously been shown to be regulated by post-translational modifications to residues on its cytoplasmic tail. A LAT knock-in strain with a tyrosine-to-phenylalanine mutation (LAT^Y136F^) peculiarly exhibits a partial block in T cell maturation combined with a lethal autoimmune and lymphoproliferative disorder [16], mimicking human IgG4-related disease [17]. Ubiquitylation of lysines K52 and K204 promotes the degradation of internalized LAT and prevents efficient recycling of LAT to the plasma membrane [18,19]. Moreover, LAT palmitoylation at cysteines 26 and 29 is required for targeting LAT from the Golgi apparatus to the plasma membrane [20]. A less studied aspect of LAT biology is, however, its redox regulation. ROS, though mostly studied for their role in host defense, have in recent times emerged as important signaling molecules that can regulate physiological processes [21], particularly through selective oxidation of the –SH group of protein cysteine residues, particularly in protein tyrosine phosphatases such as SHP-1, CD45 [22,23,24], and the protein tyrosine phosphate non-receptor type 22 (PTPN22) [25]. Changes in redox homeostasis in T cells have been shown to be associated with the pathogenesis of various diseases [26]. During our studies on NCF1 polymorphisms, we found that T cell responses were critically affected [8]. We hypothesized that the secretion of ROS into the antigen-presenting cell–T cell synapse leads to the oxidation of sensitive cysteines in the TCR signaling complex [8,27], including the LAT molecule. An interplay between ROS and LAT from studies of arthritis has been suggested; synovial fluid (SF) T cells from RA patients have been hypothesized to be hypoactive due to the redox-dependent displacement of LAT from the membrane to the cytoplasm [28], an effect possibly mediated by its cysteine residues. To study if LAT could be redox-regulated in vivo, we generated a LAT mutant mouse strain with cysteine-to-serine mutations at positions 120 and 172 (referred to as LAT^SS^). We found that the redox insensitivity of LAT affects its localization and phosphorylation, leading to changes in T cell selection, susceptibility to inflammation, and development of arthritis, an effect reversed by the NCF1 mutation.

## 2. Materials and Methods

Generation of knock-in mice expressing LAT^C120S/C172S^ molecules. A C57BL/6J mouse model (Charles River, Wilmington, MA, USA) with cysteine-to-serine mutations at positions 120 and 172 of the LAT protein was generated. A 13.7 kb genomic fragment containing exons 1 to 11 of the mouse *Lat* gene (ENSMUSG00000030742) was isolated from a BAC clone (number RP23-235A13) of C57BL/6J origin. Using ET recombination [29], the cysteine residue encoded by exon 7 and corresponding to position 120 of the LAT protein was replaced with a serine residue, and the cysteine residue encoded by exon 9 and corresponding to position 172 of the LAT protein was replaced with a serine residue. Moreover, the LAT^C120S^ mutation was associated with a silent mutation to create a diagnostic SalI restriction site, and the LAT^C172S^ was associated with a silent mutation to create a diagnostic BspeI restriction site. Finally, a self-excising ACN cassette coding for the selectable marker gene *Neo*^r^ [30] was introduced in the targeting vector 200 bp 5′ to the start of exon 9, and a cassette coding for the diphtheria toxin fragment A abutted to the 5′ end of the targeting vector [31]. JM8.F6 C57BL/6N ES cells [32] were electroporated with the targeting vector. After selection in G418, ES cell clones were screened for proper homologous recombination and the presence of the intended mutations by Southern blot and PCR analysis and subsequently injected into FVB blastocysts. Screening of the F0 mice was performed by PCR using the following pair of primers: sense 5′-GCAAGAGCTGATAACATTGAGA-3′ and antisense 5′-GAATATTCAGGAAAAGCGAGGG-3′. This pair of primers amplified a 256 bp band and a 341 bp band in the case of the WT *Lat* allele and of the *Lat^C120S/C172S^* allele, respectively. The resulting mice are denoted here as LAT^SS^ and are also known as B6-*Lat^tm8Mal^*.

Generation of knock-in mice expressing LAT^YPet^ molecules. An approach identical to the one described above for the generation of knock-in mice expressing LAT^C120S/C172S^ molecules was used to produce C57BL/6J knock-in mice that gave rise to LAT fusion proteins in which the C-terminus of LAT is fused via a Gly-Ser-Gly spacer to Ypet, a yellow fluorescent protein derived from *Aequorea victoria* (https://www.fpbase.org/protein/ypet/, accessed on 6 April 2024). The resulting mice are denoted here as LAT^SS^ and are also known as B6-*Lat^tm10Mal^*.

Other animal models. The NCF1^m1j^ mutant strain has been described extensively before [6]. All mice were backcrossed to a B10.Q strain (referred to as BQ), expressing the MHC class II A^q^ allele on a C57BL/6 background [33]. All experiments included mixing in cages, littermate controls, and blinded examinations. Mice were kept under specific pathogen-free (SPF) conditions in the animal house of the Section for Medical Inflammation Research, Karolinska Institute in Stockholm. Animals were housed in ventilated cages containing wood shavings, with a napkin as enrichment, in a climate-controlled environment with a 14-hour light-dark cycle and fed with standard chow and water ad libitum. All experimental procedures were approved by the ethical committees in Stockholm, Sweden, with the following ethical permit numbers: 12923/18 and N134/13 (genotyping and serotyping), N35/16 and 2660-2021 (DTH, CIA).

Delayed Type Hypersensitivity (DTH). Twelve-week-old mice were sensitized by intradermal injection of 100 μg rat collagen type II (COL2) in 100 μL of a 1:1 emulsion with Complete Freund’s Adjuvant (BD, Difco, Detroit, MI, USA) and 10 mM acetic acid at the base of the tail. Eight days after sensitization, the right ear was injected intradermally with 10 μL of antigen in PBS (1 mg/mL) whilst the control left ear was injected with 10 μL acetic acid in PBS. Ear swelling response was measured 0, 24, 48, and 72 h after the challenge using a caliper. Change in ear thickness was calculated by subtracting the swelling of the PBS-injected ear from the swelling of the COL2-injected ear normalized to day 0 thickness. Ear tissue was processed according to [34].

Collagen-induced arthritis (CIA). Twelve-week-old mice were immunized with 100 µg COL2 in 100 µL of a 1:1 emulsion of CFA (BD, Difco, Detroit, MI, USA) and PBS intradermally at the base of the tail. Mice were challenged on day 35 with 50 µg COL2 in 50 µL of an IFA/PBS (BD, Difco, Detroit, MI, USA) emulsion. Mice were monitored for arthritis development every second to third day using the following scoring system: each visibly inflamed (i.e., swelling, redness) ankle or wrist was given 5 points, whereas inflamed knuckles were given 1 point, resulting in a total of 60 possible points per mouse.

Genomic DNA isolation and genotyping. Toes were boiled at 90 °C for 90 min in 50 µL of 0.5 mM NaOH. Lysates were neutralized with 50 µL Tris-HCl. Then, 2 µL of the lysate was used for subsequent PCR. Genotyping PCR products were diluted 1:12 for microsatellite typing on an ABI-3730 machine. SNP typing was performed by high-resolution melting using iQ SYBR Green (BioRad, Hercules, CA, USA) qPCR mix.

Preparation of single-cell suspension from lymphoid organs. Spleens or lymph nodes were mashed using a 1 mL syringe plunger on a 40 µM cell strainer (Falcon). The cell suspension was washed in PBS and centrifuged at 350× *g* for 5 min at room temperature (RT). For spleens, red blood cells were lysed in 1 mL of ammonium–chloride–potassium (ACK) lysis buffer (155 mM NH_4_Cl, 12 mM NaHCO_3_, 0.1 mM EDTA, homemade) for 1–2 min at RT. Thereafter, cells were washed in PBS and carefully transferred to a new 15 mL tube to get rid of debris. Cells were centrifuged and resuspended in 1–3 mL of DMEM for counting in a Sysmex KX-21N cell counter.

CD4 T cell isolation and in vivo transfer. CD4 T cells were enriched with a kit following the manufacturer’s instructions (untouched CD4 T cell Mouse Dynabead kit, Life Technologies, Carlsbad, CA, USA). A total of 1 × 10^6^ CD4 T cells from BQ or LAT^SS^ were injected intravenously in TCRβ knock-out mice and 4 days later, DTH was performed.

Confocal microscopy. First, 0.2 × 10^6^ splenocytes were stained with surface anti-mouse CD3 (17A2 eBioscience, Waltham, MA, USA, 1:100) in FACS buffer (2% FCS, 2 mM EDTA in PBS) for 20 min at RT, washed, and fixed with Cytofix/Cytoperm (BD, Franklin Lakes, NJ, USA) for 20 min at 4C. Cells were then stained intracellularly with anti-LAT (CST #45533; 1:100) for 20 min at RT, washed, and then stained with goat anti-rabbit AF647 (A-21245 Invitrogen, Carlsbad, CA, USA, 1:500). Nuclei were stained using DAPI at 0.5 μg/mL for 10 min in PBS at RT. After staining, cells were resuspended in PBS at a concentration of 10^6^ cells/mL, and 10 μL of the cell suspension was plated on microscopy slides (Thermo Fisher Scientific, Waltham, MA, USA). Slides were left to dry in the dark at RT for 1 h and mounted using Vectashield mounting medium (Vector Laboratories, Newark, CA, USA). Slides were imaged on a Zeiss LSM 800 confocal laser scanning microscope. Image analysis was performed using ImageJ 1.52i.

Cell culture. A total of 1 × 10^6^ splenocytes or 5 × 10^5^ lymph node cells were cultured in 200 µL of complete DMEM per well in U-shaped bottom 96-well plates (Nunclon, Waltham, MA, USA). Cells were incubated at 37 °C and 5% CO_2_ without using the outermost wells to avoid artifacts related to evaporation. Complete DMEM: DMEM with Glutamax (Gibco, Waltham, MA, USA); 5% FCS (Gibco); 10 µM HEPES (Sigma, Burlington, MA, USA); 50 µg/mL streptomycin sulfate (Sigma); 60 µg/mL penicillin C (Sigma, Burlington, MA, USA); and 50 µM beta-mercaptoethanol (Gibco). FCS was heat-inactivated for 30 min at 56 °C in a water bath, aliquoted, and kept frozen at −20 °C until further use. The following stimuli were used: H_2_O_2_ (20 µM, Sigma), BSO (200 µM, Sigma), galCOL2_260–270_ (10 μg/mL, in-house), anti-mouse CD3 (1 μg/mL, 145-2C11, BD), anti-mouse CD28 (1 μg/mL, 37.51, BD), and Phorbol 12-Myristate 13-Acetate (PMA, 20 ng/mL). For proliferation and immunoblot assays, lymph node cells or enriched CD4 T cells (untouched CD4 T cell Mouse Dynabead kit, Life Technologies) were plated on pre-coated (10 µg/mL anti-CD3 and 5 µg/mL anti-CD28) 96-well U bottom plates (Costar, medium binding) overnight at 4 °C or 2 h at 37 °C in PBS.

Flow cytometry. All centrifugation steps were carried out at 350× *g* for 5 min at RT. A total of 1 × 10^6^ cells were blocked in 20 µL PBS containing 5 µg in-house 2.4G2 in 96-well plates for 10 min at RT. Samples were washed in 150 µL PBS and subsequently stained with the indicated antibodies in 20 µL PBS diluted 1:100 or 1:200 at 4 °C for 20 min in the dark. Cells were washed once and fixed–permeabilized for intracellular staining using BD Cytofix/Cytoperm™ (BD, Franklin Lakes, NJ, USA) according to the manufacturer’s instructions. Cells were stained intracellularly in 20 µL permeabilization buffer (BD, Franklin Lakes, NJ, USA), using the antibodies at a 1:100 final dilution, for 20 min at RT. Foxp3 staining required nuclear permeabilization and was carried out using Bioscience™ Foxp3/Transcription Factor Staining Buffer instead. For intracellular cytokine staining, cells were stimulated in vitro with PMA 10 ng/mL, ionomycin 1 µg/mL, and BFA 10 µg/mL for 4–6 h at 37 °C before fixation, permeabilization, and staining. Dead cells were excluded using a fixable near-IR dead cell stain kit (Thermo Fisher Scientific, Waltham, MA, USA, catalog number L10119). Samples were acquired with Attune NxT flow cytometer (Thermo Fisher Scientific) and analyzed with FlowJo version 10.8.

Flow cytometry antibodies: Antibodies were purchased from BD unless stated otherwise. Antibodies are listed by recognized antigen, and the clone is indicated in parenthesis, as follows: CD3 (145-2C11), TCRβ (H57-597), CD4 (RM4-5), CD8 (53-6.7), Foxp3 (FJK-16s), CD25 (7D4), CD44 (IM7), CD62L (MEL-14), and IFN-γ (R46A2).

Calcium flux. Lymph node and spleen cells were stained in pre-warmed PBS + 1% FCS with Fluo4-AM (2 µM, Thermo Fisher) and FuraRed AM (4 µM, Thermo Fisher) at 37 °C. Cells were washed in cold PBS + 1% FCS before staining for extracellular markers for 20 min. Baseline calcium flux was recorded for 100 s before 50 µL anti-CD3 (10 µg/mL; BD) stimulation was added. Five minutes later, the maximum flux was measured using ionomycin (1 µg/mL; BD). Relative calcium concentration was plotted as a ratio of Fluo3 to FuraRed emission using FlowJo.

Proliferation assay. For in vitro proliferation, 1 × 10^7^ lymph node or spleen cells per mouse were labeled using the CellTrace™ Violet Cell Proliferation Kit (Thermo Fisher) according to the manufacturer’s instructions. A total of 5 × 10^5^ labeled cells were plated per well (96-well U bottom plate, Nunclon) and stimulated with anti-mouse CD3 (1 µg/mL, 500A2, BD Pharmingen) and anti-mouse CD28 (1 µg/mL, 37.51, BD) for 72–96 h. Proliferation by dilution of CTV was assessed using flow cytometry. Proliferation parameters were analyzed and calculated using the Proliferation Tool in FlowJo 8.8.7.

Total protein isolation. Total protein was isolated from 2 × 10^6^ lymph node cells stimulated with anti-CD3/CD28 1 µg/mL for 0, 2, and 5 min. Cells were lysed in 30 µL lysis buffer (M-Per; ThermoFisher) with freshly added protease inhibitors (Halt 100×, ThermoFisher), heated to 95 °C for 15 min, thoroughly mixed, and heated again for 10 min at 95 °C. Lysates were centrifuged in a table centrifuge at top speed for 15 min, and supernatants were transferred to a new tube and kept at −20 °C until further use. Protein concentration was measured using NanoDrop 2000 and then adjusted.

SDS-PAGE. First, 40 µg of total protein was diluted in NuPAGE LDS buffer containing beta-mercaptoethanol and then loaded onto a 4-12 NuPAGE Bis-Tris gel (Thermo Scientific). The gel ran at RT for 45 min at 200 V using NuPAGE MOPS running buffer.

Western blot. Proteins were blotted onto a PVDF membrane (Millipore, Burlington, MA, USA) for 1.5 h at 35 V in NuPAGE transfer buffer (Thermo Fisher) containing 20% methanol at RT. Membranes were blocked for 1 h at RT in PBS containing 0.05% tween-20 (PBS-T) and 5% BSA (VWR). Incubation with the primary antibody was performed overnight at 4 °C in a blocking solution. β-Actin was used as a loading control (Abcam). After incubation, the membrane was washed 3 times, each time for 5 min, in PBS-T at RT. The membrane was incubated with the secondary antibody (Affinipure peroxidase-coupled goat anti-mouse IgG (H + L), final conc. 40 ng/mL, Jackson laboratories) for 1 h at RT and washed. The membrane was incubated with either homemade ECL (solution A (final conc. 0.1 mM Tris-HCl, pH 8.0; 400 µM coumaric acid; 2.5 mM luminol) and solution B (final conc. 0.1 mM Tris-HCl, pH 8.5; 0.03% H_2_O_2_) or by using a commercial ECL substrate solution (GE Healthcare, Chicago, IL, USA) before imaging on a ChemiDoc XRS+ (BioRad, Hercules, CA, USA).

ELISA. Mice were bled from the sub-mandibular vein at the indicated time points. Serum was harvested and used in ELISA to detect anti-COL2 antibodies. Briefly, rCOL2, used for immunization, was diluted in PBS to a final concentration of 10 µg/mL, and 50 µL was added to each well to a 96-well MaxiSorp plate and incubated overnight at 4 °C. Plates were blocked with 1% BSA for 1 h at room temperature. Serum samples typically diluted 1:1000 or 1:10,000 in PBS were incubated at room temperature for 2 h. Detection antibodies HRP-coupled were diluted 1:4000 (Southern Biotech, Birmingham, AL, USA) in PBS, and samples were developed either with ABTS reading the absorbance at 405 nm or TMB reading the absorbance at 450 nm.

Bead-Based Multiplex Immunoassay. Antibody responses were analyzed by using the Luminex platform as described previously. At first, Neutravidin (Thermo Fischer Scientific) was coupled to magnetic carboxylated beads (Magplex Microspheres, Luminex Corp, Austin, TX, USA), after which biotinylated COL2 peptides were immobilized to the beads. Human and mice serum samples were diluted 1:100 (*v*/*v*), and purified antibodies were diluted into an assay buffer (3% BSA, 5% milk powder, 0.05% Tween 20, 100 μg/mL Neutravidin in PBS) and incubated for 1 h at RT on a shaker. The samples were then transferred to a 96-well plate (Greiner Bio-One, Kremsmünster, Austria) containing the peptide-coated beads. After incubation at RT on a shaker for 75 min, all the beads were washed with 0.05% Tween-20 in PBS (PBST) on a plate washer (Bioplex Pro Wash station, BioRad) and then resuspended in a solution (PBST, 3% BSA) containing the secondary anti-human or anti-mouse IgG Fcγ-PE (Jackson Immuno Research, West Grove, PA, USA). After 40 min of incubation, the beads were washed with PBST, and the fluorescence intensity was measured using a Bioplex 200 instrument (BioRad). The median fluorescence intensity (MFI) was used to quantify the interactions of the antibody with the given peptides for monoclonal antibodies and mice sera. For the comparison of responses to peptides in human serum samples, the ratio value, calculated by dividing the MFI value for the peptide of interest by the MFI value of either cyclic peptides or the THP control peptide, was used.

Statistical analysis. Graphs were plotted as mean ± SD and statistical analysis was done with GraphPad Prism version 9.0. Mann–Whitney test was used when comparing two groups and 2-way ANOVA was used when comparing multiple variables. The significance level was set to 0.05, and the *p*-values are indicated with asterisks (* *p* < 0.05, ** *p* < 0.01, *** *p* < 0.001, and **** *p* < 0.0001).

## 3. Results

### 3.1. LAT Phosphorylation Is NOX2-Dependent

To determine if LAT (ENSMUSG00000030742) is redox-regulated, we first set out to investigate whether LAT localization is dependent on the redox state of a T cell, as previously suggested by Gringhuis [35]. For that, we created a mouse model where the C-terminus of LAT was fused to YPet, a yellow fluorescent reporter protein (LAT-YPet). Treatment of LAT-YPet cells with H_2_O_2_ resulted in a significant reduction in LAT fluorescent signal in the membrane (Figure 1A and Figure A1), showing that an oxidative microenvironment affects intracellular LAT localization. As the majority of inducible ROS in cells derives from the NADPH oxidase 2 (NOX2 complex), we decided to use a mouse model that harbors a natural mutation in the *Ncf1* gene [6], part of NOX2. This mutation generates a truncated NCF1 protein that is incapable of interacting with *Ncf2*, therefore affecting the functionality of the NOX2 complex and resulting in a lower oxidase burst in all cell types [1]. To investigate how LAT is controlled by ROS, we created a mouse model with a cysteine-to-serine mutation at positions 120 and 172 (LAT^SS^) (Figure A2A,B). Under oxidative conditions in vitro, these serine residues had previously been hypothesized to form disulfide bonds with cysteines 26 and 29, which are proximal to the LAT α-helix that is inserted into the plasma membrane [35]. Therefore, mutating them would render LAT insensitive to possible redox changes (schematic in Figure 1B). Treatment of human T cells with BSO (L-buthionine-S, R-sulfoximine), which enhances ROS pressure by decreasing intracellular levels of the key antioxidant glutathione, has previously been shown to affect redox-dependent changes in T cell responsiveness [36]. In wild-type cells, BSO treatment enhanced the intracellular localization of LAT when compared with untreated cells. In LAT^SS^ cells, the increased oxidative pressure had no impact on protein localization with LAT primarily detected on the cell surface, confirming that the LAT^SS^ protein is indeed redox-insensitive (Figure 1C). However, the LAT^SS^ mutation still increased LAT phosphorylation on the NCF1-NOX2-ROS-deficient background (Figure 1D).

### 3.2. LAT^SS^ Impacts Thymic Selection and Peripheral T Cells at Steady State

Given the importance of LAT in TCR signaling, we wanted to investigate whether increased LAT activity in the naive LAT^SS^ mice affected T cell selection. For that, we phenotyped thymi, lymph nodes, and blood using flow cytometry. We found that thymi from the LAT^SS^ mice had lower frequencies of double-positive CD4+ CD8+ (DP) thymocytes, which expressed higher levels of CD69, compared with thymi from wild-type littermates (Figure 2A,B). This is of particular interest as LAT expression is highest in this population [37]. In contrast, we found no difference in the double-negative CD4- CD8- (DN) population identified by CD44 and CD25 staining (Figure 2A,C). In peripheral lymphoid organs, we did not detect any differences in overall CD4+ T cells, activated CD4+ T cells (CD69+), or regulatory T cells (Tregs, Foxp3+ CD25+) (Figure 2D). However, we observed a decrease in naive CD4+ CD62L+ cells and a concomitant increase in CD4+ CD62L- CD44- T cells in the LAT^SS^ mouse (Figure 2E). In the blood, we found an expansion of Vβ8.1+ T cells, a TCR that has been implicated in arthritis pathogenesis [38] (Figure 2F). Together with our in vitro studies, these results suggest that naive LAT^SS^ T cells have higher basal activity.

To investigate T cell function in more detail, we determined calcium influx, one of the earliest signaling events upon TCR engagement. After stimulation with anti-CD3, we observed no differences in calcium influx levels between LAT^SS^ and wild-type littermates (Figure A3A). Since T cells express numerous calcium-permeable channels at various locations, we hypothesize that the LAT^SS^ mutation alone does not affect the calcium influx. To understand whether the LAT mutation affected T cell proliferation, we assessed by CellTrace dilution T cell proliferation. After stimulation with anti-CD3/CD28, we observed no differences between the groups, concluding that the mutation in the LAT molecule does not affect the proliferation or activation of peripheral T cells, as observed in the thymus (Figure A3B,C). Finally, we investigated whether other TCR-related signaling proteins were affected by the LAT mutation. After polyclonal TCR stimulation of naive CD4 T cells in vitro, we did not find differences in the phosphorylation of the TCR-related signaling proteins ERK, FYN, LCK, PKC, and ZAP70 (Figure A3D).

### 3.3. LAT^SS^ Enhances T Cell-Dependent Inflammation in DTH and CIA Models

To address the importance of redox regulation of T cells through LAT in vivo, we used a Th1-driven delayed-type hypersensitivity (DTH) model [39]. Mice were immunized with rat collagen type 2 (COL2), which is known to activate T cells against the COL2_260-270_ peptide, galactosylated at position 264 [40]. We observed increased ear swelling in mutant mice when compared with wild-type littermates, with PBS-injected ears counted as baseline (Figure 3A). This could be explained by an observed increased infiltration of CD45+ cells into the ears of LAT^SS^ mice as well as increased IFN-γ production by CD44+ T cells in lymph nodes after ex vivo stimulation (Figure 3B,C). To corroborate that the observed phenotype was mediated by T cells, we repeated the experiment in TCRβ knock-out (TCRβ-KO) mice reconstituted with wild-type BQ or LAT^SS^ T cells. As shown in Figure 3D, mice that received CD4 T cells from the LAT^SS^ background developed more severe ear inflammation as compared with the control animals, confirming that LAT insensitivity of LAT enhances T cell-dependent inflammation. Moreover, we checked the blood 12 days after CD4 T cell transfer as well as IFN-γ production by ear lymph nodes 48 h post-DTH and observed increased IFN-γ in the LAT^SS^ mice (Figure A4A,B).

We next investigated whether the absence of induced ROS through the NCF1^m1j^ mutation would affect a T cell-dependent disease model. For that, we tested the collagen-induced arthritis (CIA) model, a commonly used model that recapitulates many features of human RA. Indeed, LAT^SS^ mice developed an earlier onset of disease with higher incidence until day 31 (Figure 4A), and a reversed effect was observed under NCF1-ROS-deficient conditions, with a slower disease progression in LAT^SS^/NCF1^m1j^ mice (Figure 4B). Moreover, in vitro T cell recall assays showed increased production of IL-2 and IFN-γ in the LAT^SS^ mice (Figure 4C). In the spleen, LAT^SS^ mice had fewer CD4 T cells with higher expression of CD40L, indicating that these T cells are more active (Figure 4D). However, we observed no difference in the NCF1^m1j^-deficient background (Figure 4D). Overall serum antibody levels against COL2 were comparable among groups (Figure 4E). Interestingly, when investigating the fine specificity of the antibody response using a Luminex assay, we could surprisingly see a clear increase in antibody reactivity to the F4 epitope of COL2 in the LAT^SS^ mice, both under ROS-sufficient and -deficient conditions (Figure 4F). Antibodies against F4 have been shown to ameliorate arthritis severity [41] and could be a regulatory feedback mechanism to the loss of control induced by the oxidation-insensitive mutant LAT. Taken together, our data demonstrate that LAT has redox sensing capabilities and acts to moderate TCR signaling and T cell activation in the presence of ROS from the NCF1/NOX2 complex.

## 4. Discussion

The view of ROS has over the past decades shifted, no longer focusing on their deleterious effects, but rather appreciating their subtle role in modulating physiological pathways and acting as a second messenger [42,43,44,45]. T cells are subject to significant ROS pressure because of both enhanced T cell metabolism and the presence of ROS-producing innate immune cells [1,8,46,47]. MAPK signaling in T cells, which mediates proliferation and survival, has been reported to be redox-sensitive [48], and mitochondrial complex III-derived ROS have been shown to be important for the expansion of antigen-specific CD4 T cells [49]. We have previously shown that ROS deficiency through mutations in the NCF1 protein, an essential component of the NOX2 complex in APCs, increases susceptibility to arthritis, systemic lupus erythematosus (SLE), and autoimmune encephalomyelitis [1,2]. Moreover, surface thiol levels on T cells determine T cell reactivity and arthritis susceptibility [8]. Several target cysteines, in the same or interacting cells, are likely regulated by ROS from the NCF1/NOX2 complex, and earlier, we have shown cysteine regulation of PTPN22 in APCs [25]. However, the consequences of redox regulation of key signaling proteins in inflammation remain poorly understood.

Here, we studied LAT, an essential molecule that couples T cell receptor activation to downstream T cell signaling pathways, and how it is redox-regulated. Analysis of oxidized LAT under reducing and non-reducing conditions revealed the presence of one or more disulfide bonds, and cysteine 117 of human LAT was shown to be redox-sensitive [36]. Therefore, to study the redox regulation of LAT, we used an in vivo approach of mutating both C120 and C172 to serines preventing their regulation by redox levels. In contrast to wild-type LAT, an oxidative microenvironment did not affect LAT^SS^ localization on the membrane. Therefore, in line with Gringhuis’ data [28], we may hypothesize that under oxidative conditions, the SH group of wild-type C120/C172 in the mouse forms a disulfide bond with the SH groups of C26/C29. As the latter are either part of (C26) or proximal to (C29) the alpha-helix, the resulting conformational change may cause sterical hindrance, preventing proper incorporation of the protein into the plasma membrane [36].

In T lymphocytes, LAT is present both at the plasma membrane and in intracellular vesicles and is dynamically shuttled to and from the immunological synapse in a Rab6- and Synatxin-16-dependent fashion [18,50,51]. Here, we observed less LAT in the membrane after H_2_O_2_ treatment; however, further experiments are needed to precisely address how a shift in the redox milieu affects LAT, be it through changes in internalization, recycling, or degradation, and whether these are a result of altered structural integrity. Structural studies of LAT are complicated by its intrinsically disordered regions, characteristic of proteins that act as signaling hubs [52,53]. While the inherently unfolded nature allows them greater flexibility and accessibility, they hinder the structural investigation of the protein.

LAT is critical for thymic selection, but its relation to ROS and how this may affect thymic selection have not been explored. LAT^−/−^ mice display a block in thymocyte development at the CD25+ CD44- DN3 stage with a lack of mature T cells in the periphery [14], highlighting the role of LAT in pre-TCR signaling, and deletion of LAT at the late DN3 stage results in DP cells with abrogated TCR signaling [15]. Furthermore, THEMIS (Thymocyte-expressed molecule involved in selection), a major regulator of positive selection and a participant in the LAT signalosome, has been shown to regulate the redox state of SHP-1 in response to physiological ROS [54]. Here, we show that redox insensitivity of LAT leads to a marked reduction in CD4+ CD8+ T cells in the thymus, reminiscent of the phenotypes observed in thymocytes deficient in other LAT signalosome molecules such as LAT, LCK, or SLP-76 [55,56,57]. Moreover, we have also observed decreased CD62L in T cells from the LAT^SS^ mice. CD62L, also known as L-selectin, is a cell adhesion molecule expressed on the surface of naive T cells, which plays a crucial role in lymphocyte trafficking and homing to secondary lymphoid organs. This observation, together with the expansion of Vβ8.1+ T cells, a TCR that has been implicated in arthritis pathogenesis [38], led us to hypothesize that the mutation in the LAT molecule promotes the higher basal activity of these T cells, which might in turn lead to increased autoimmunity as we observed in the CIA model. Thus, together with our previous data [25], we now demonstrate that the redox sensing by the TCR signalosome is critical for thymic selection with the involvement of both LAT in T cells and PTPN22 in APCs [25]. Upon TCR engagement by APCs, LAT becomes phosphorylated and serves as a scaffold for various signaling proteins, leading to downstream signaling events that ultimately result in T cell activation, proliferation, and effector functions. On the other hand, PTPN22 negatively regulates TCR signaling and immune responses by dephosphorylating key signaling proteins involved in T cell activation, thus serving as a negative regulator of T cell activation. The interplay between LAT signaling in T cells and PTPN22-mediated regulation in APCs helps fine-tune the immune response, ensuring an appropriate level of T cell activation and preventing excessive or dysregulated immune responses.

When studying the effects of LAT^SS^ on inflammation and arthritis, we observed enhanced inflammation in the DTH model and increased susceptibility in the CIA model. Moreover, we also observed increased IL-2 and IFN-γ responses, accompanied by decreased CD4 T cells but with higher expression of CD40L. These T cells heightened the expression of CD40L probably to compensate for their numerical scarcity by more effectively engaging and stimulating APCs. Th1-driven inflammation could be transferred by CD4 T cells, showing that redox insensitivity renders LAT^SS^ T cells more active.

Considering LAT’s crucial function in potentiating T cell activation, it stands to reason that surface levels of LAT need to be tightly controlled. Therefore, sensitivity to redox changes may in addition to ubiquitylation and palmitoylation represent an additional mechanism of LAT regulation. Taken together, our experiments indicate that LAT is redox-regulated, interacting with NCF1/NOX2 complexes, and may therefore be a key target for understanding and intervening in the treatment of RA and other autoimmune diseases.

## 5. Conclusions

LAT is a key T cell protein. Here, we show that LAT is redox-regulated, having an impact on thymic selection, peripheral T cell populations, and autoimmunity. This study provides further insight into LAT regulation, thereby altering downstream signaling and inflammatory responses.

## Figures and Tables

**Figure 1 antioxidants-13-00499-f001:**
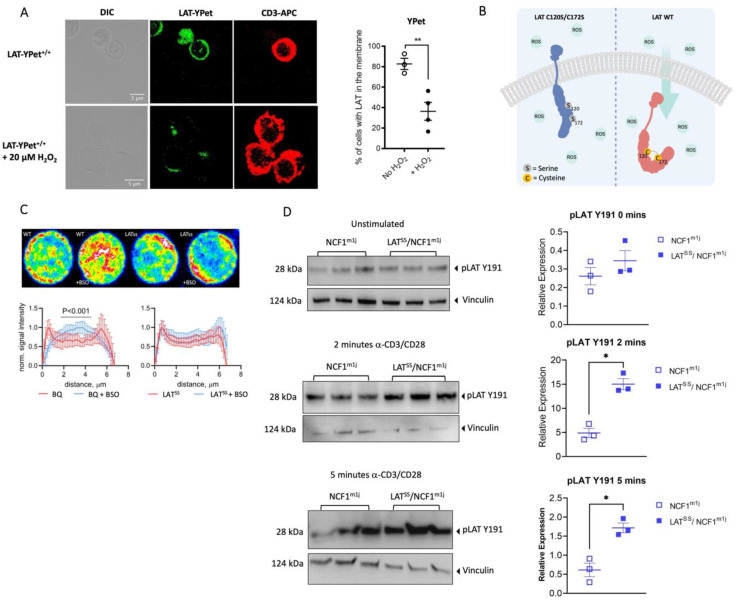
Cysteines 120 and 172 mediate LAT activation through redox regulation. (**A**) Confocal differential interference contrast (DIC) and fluorescence images of LAT-YPet cells with and without treatment with 20 μM H_2_O_2_. (**B**) Hypothetical model for LAT displacement in oxidative conditions: (left) the sulfhydryl group of C120/C172 forms a disulfide bond with the sulfhydryl group on C26/C29 causing sterical hindrance and thereby displacement from the membrane; (right) cysteine-to-serine mutations render LAT insensitive to redox regulation. (**C**) Confocal microscopy image showing LAT distribution (in red) in WT and LAT^SS^ lymph node cells either untreated or treated with the glutathione inhibitor BSO. (**D**) LAT Y191 phosphorylation in unstimulated lymph node cells, and 2 and 5 min upon stimulation with anti-CD3/anti-CD28. Each lane represents an individual mouse. On the right, quantification shows phosphorylated protein expression normalized to vinculin loading control. Error bars represent mean ± SEM. * *p* < 0.05, and ** *p* < 0.01.

**Figure 2 antioxidants-13-00499-f002:**
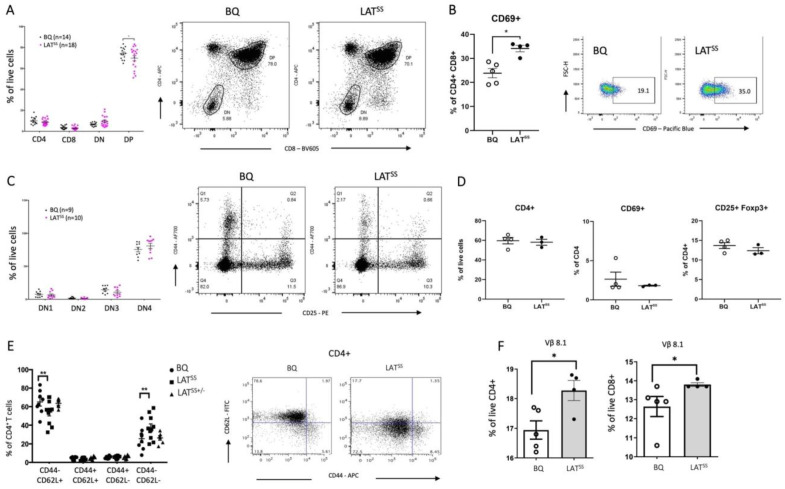
LAT^SS^ affects thymic selection and peripheral T cell populations. (**A**–**C**) Flow cytometry analysis of T cell populations in the thymus of littermate BQ (wild-type) and LAT^SS^ mice with representative flow cytometry plots shown; DN1 (CD44+ CD25−), DN2 (CD44+ CD25+), DN3 (CD25+ CD44−), DN4 (CD25− CD44−). (**D**) Percentages of CD4+, CD4+ CD69+, and CD4+ CD25+ Foxp3+ T cells in naive lymph nodes. (**E**) CD44 and CD62L expression on CD4 T cells in naive lymph nodes. Representative gating is shown. (**F**) Vβ chain usage on naive CD4/CD8 T cells in the blood measured by flow cytometry. Error bars represent mean ± SEM. * *p* < 0.05, and ** *p* < 0.01.

**Figure 3 antioxidants-13-00499-f003:**
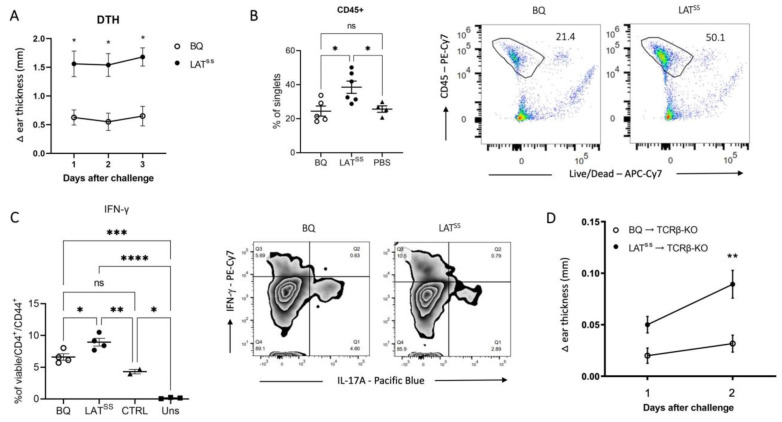
LAT^SS^ enhances inflammation in the DTH model. (**A**–**C**) Mice were immunized according to the DTH protocol with COL2. (**A**) Mean Δ ear pinna thickness in mm (calculated as COL2-injected ear swelling minus PBS-injected ear swelling) 24, 48, and 72 h after COL2 challenge; BQ (*n* = 4), LAT^SS^ (*n* = 5). Data are representative of 3 experiments. Two-way ANOVA. (**B**) Percentage of CD45+ cells in wild-type BQ and LAT^SS^ COL2-injected ear compared to PBS-injected ear measured by flow cytometry and representative gating on the right. (**C**) Intracellular flow cytometry staining of IFN-γ in inguinal lymph nodes after stimulation with PMA and representative gating on the right. (**D**) Mean Δ ear pinna thickness in mm in TCRβ knock-out mice post-transfer of purified CD4+ T cells from BQ and LAT^SS^ mice 10 days after DTH induction. Error bars represent mean ± SEM. * *p* < 0.05, ** *p* < 0.01, *** *p* < 0.001, and **** *p* < 0.0001.

**Figure 4 antioxidants-13-00499-f004:**
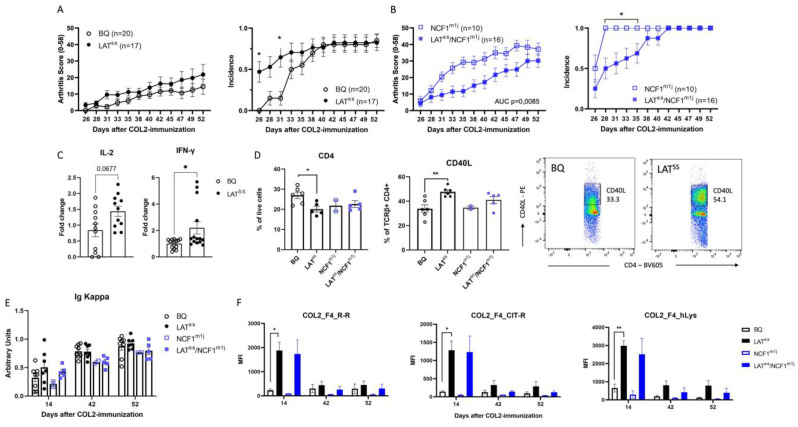
LAT^SS^ regulates CIA susceptibility in an NCF1-dependent manner. (**A**–**E**) Mice were immunized according to the CIA protocol with COL2. (**A**,**B**) Clinical score and incidence of littermate mice immunized with COL2. Data are representative of 3 independent experiments. (**C**) IL–2 and IFN-γ production by T cell recall assay. (**D**) Percentage of CD4+ and CD4+ CD40L+ T cells in draining lymph nodes measured by flow cytometry. Representative gating for CD40L is shown. Error bars represent mean ± SEM. (**E**) Levels of serum antibodies against COL2 measured by ELISA. (**F**) Luminex analysis of serum antibody reactivity against F4 epitope on COL2; wild-type peptide R-R with unmodified arginine residues, Cit-R with citrullinated arginine residues, and hLys with hydroxylated lysine modifications. * *p* < 0.05, and ** *p* < 0.01.

## Data Availability

Data are contained within the article.

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
