# Peer review of "Redox Regulation of LAT Enhances T Cell-Mediated Inflammation"

_antioxidants, 2024, doi:10.3390/antiox13040499_

Round 1

Reviewer 1 Report

By generating a mouse strain with redox-sensitive cysteine mutations within LAT, the authors further observed the redox regulation of LAT through C120 and C172 mediated LAT membrane localization and phosphorylation in T cells, and tried to address the possibility that ROS regulate T cell-dependent inflammation through LAT. Interestingly, they showed that redox insensitivity of LAT leads to a reduction in DP T cells in the thymus and reduced CD62L+naïve CD4 T cells in periphery.  Importantly, redox insensitivity of LAT enhanced inflammation in the DTH model and collagen induced arthritis which is in an NCF1-dependent manner. However, this reviewer felt some results had no sufficient interpretations. e.g.  How and why H2O2 oxidation resulted in reduction of LAT in the membrane? Internalized and degraded? or why peripheral CD62L+naïve CD4 T cells are reduced in the naive LATSS mice? Why naive LATSS T cells have higher basal activity? Why there is an expansion of V8.1+T Cells in blood of naive LATSS mice in steady state? What scenario would connect LAT in T cells and PTPN22 in APCs?  Why in the spleen, LATSS mice had less CD4 T cells with higher expression of CD40L?..

    Most experiments have been designed and performed in suitable ways. There are several concerns in the present form of this manuscript.

Figure 1A: why only one CD3+ T cell shown instead of 3 LAT-YPet T cells in right upper panel?

Figure 1D: in the WB of p-LAT in LN cells (CD4, CD8 Or mix?) with 2 mins anti-CD3/CD28 stimulation, the “medium” without stimulation is missing. The loading control bands (vinculin) are very weak, need to be replaced with total LAT (the same for FigS1). This reviewer would suggest to isolated either CD4 or CD8 for WB and might have more significant effect. since these two T cell subsets may have different signaling (even opposite) responses.

Author Response

Please see the word attachment.  

Reviewer 2 Report

In this study, James et al. investigate the adaptor molecule linker for activation (LAT) of T cells (LAT) in terms of redox regulation. In general, the authors provide rational and solid evidence considering the in vitro and in vivo capacities of T cells in this setting by using a sufficient spectrum of techniques and genetic modifications.

From the experimental site some control experiments and corrections considering Figure 1 have to be made (see major experimental points).

Overall the manuscript is excellently written and understandable. The interpretations of the data were accurate and reasonable.

If the points mentioned in detail below can be addressed by the authors, this article is ready for publication and will be an interesting contribution to the field.

Major experimental points:

·         While all other Figures provide solid evidence and appropriate controls, only Figure 1, which depicts the interplay of ROS and LAT has to be improved. The most important point, which has to be addressed are some ROS measurements (with 6-carboxy-DCF) to show that the treatment with BSO in Figure 1C truly increases the cellular ROS levels in T cells. On the other hand ROS measurements have to be performed with T cells isolated from the two mouse lines NCF1m1J (to prove that there is a deficit in ROS levels) and NCF1m1J/LATSS) (to show if and which influence the two mutations have on the ROS levels) (Figure 1D). Of course always in comparison to the appropriate WT strains.

·         Also in Figure 1D. It is very nice that quantifications of the lanes were performed. However, is there not a better loading control other than Vinculin. It seems rather weak and fluctuating.

Author Response

Please see the word attachment. 

Round 2

Reviewer 1 Report

This reviewer strongly disagrees with the authors.  

1. In previous review, the questions listed "However, this reviewer felt some results had no sufficient interpretations. e.g.  How and why H2O2 oxidation resulted in reduction of LAT in the membrane? Internalized and degraded? or why peripheral CD62L+naïve CD4 T cells are reduced in the naive LATSS mice? Why naive LATSS T cells have higher basal activity? Why there is an expansion of V8.1+T Cells in blood of naive LATSS mice in steady state? What scenario would connect LAT in T cells and PTPN22 in APCs?  Why in the spleen, LATSS mice had less CD4 T cells with higher expression of CD40L?" need to be discussed in "Discussion" and show to the readers not to the reviewer.

2. The current immune blotting data presented in main and supplemental figures do not meet basic requirements. in Fig1D, 2 min stimulation? (need to be indicated in the figure), and the non-stimulated blots, even the 5 min stimulated (FigS3) can be moved to main Fig1D, making clear signaling events. 

 1. The current immune blotting data presented in main and supplemental figures do not meet basic requirements. in Fig1D, 2 min stimulation? (need to be indicated in the figure), and the non-stimulated blots, even the 5 min stimulated (FigS3) can be moved to main Fig1D, making clear signaling events.

2. In Fig 1A, the author claimed   "Treatment of LAT-YPet cells with H2O2 resulted in a significant reduction of LAT fluorescent signal in the membrane (Figure 1A), but from the authors provided additional raw images (left bottom panel), it looks opposite showing enhanced LAT (green) in H2O2 treated cells (more contrast), and the presented panel does not correspond to the "more contrast" image. 

Reviewer 2 Report

Thank you for clarification of my concerns. For this article and its main conclusions this is sufficient.

Thank you for clarification of my concerns. For this article and its main conclusions this is sufficient.

Author Response

We thank the reviewer for accepting our clarification of the presented concerns. 

Round 3

Reviewer 1 Report

This reviewer has no further comments and accepts in present form.

Fonts of each figure need to be consistent size and visible.